# Genome-Wide Analyses and Prediction of Resistance to MLN in Large Tropical Maize Germplasm

**DOI:** 10.3390/genes11010016

**Published:** 2019-12-23

**Authors:** Christine Nyaga, Manje Gowda, Yoseph Beyene, Wilson T. Muriithi, Dan Makumbi, Michael S. Olsen, L. M. Suresh, Jumbo M. Bright, Biswanath Das, Boddupalli M. Prasanna

**Affiliations:** 1Department of Agricultural Science and Technology, Kenyatta University, Nairobi 43844-00100, Kenya; christinenyaga96@gmail.com (C.N.); wtmuriithi@gmail.com (W.T.M.); 2International Maize and Wheat Improvement Centre (CIMMYT), World Agroforestry Centre (ICRAF), United Nations Avenue, Gigiri, Nairobi 1041-00621, Kenya; Y.Beyene@cgiar.org (Y.B.); d.makumbi@cgiar.org (D.M.); M.Olsen@cgiar.org (M.S.O.); l.m.suresh@cgiar.org (L.M.S.); b.jumbo@cgiar.org (J.M.B.); b.das@cgiar.org (B.D.); b.m.prasanna@cgiar.org (B.M.P.)

**Keywords:** GWAS, GP, validation, markers, resistance, maize lethal necrosis

## Abstract

Maize lethal necrosis (MLN), caused by co-infection of maize chlorotic mottle virus and sugarcane mosaic virus, can lead up to 100% yield loss. Identification and validation of genomic regions can facilitate marker assisted breeding for resistance to MLN. Our objectives were to identify marker-trait associations using genome wide association study and assess the potential of genomic prediction for MLN resistance in a large panel of diverse maize lines. A set of 1400 diverse maize tropical inbred lines were evaluated for their response to MLN under artificial inoculation by measuring disease severity or incidence and area under disease progress curve (AUDPC). All lines were genotyped with genotyping by sequencing (GBS) SNPs. The phenotypic variation was significant for all traits and the heritability estimates were moderate to high. GWAS revealed 32 significantly associated SNPs for MLN resistance (at *p* < 1.0 × 10^−6^). For disease severity, these significantly associated SNPs individually explained 3–5% of the total phenotypic variance, whereas for AUDPC they explained 3–12% of the total proportion of phenotypic variance. Most of significant SNPs were consistent with the previous studies and assists to validate and fine map the big quantitative trait locus (QTL) regions into few markers’ specific regions. A set of putative candidate genes associated with the significant markers were identified and their functions revealed to be directly or indirectly involved in plant defense responses. Genomic prediction revealed reasonable prediction accuracies. The prediction accuracies significantly increased with increasing marker densities and training population size. These results support that MLN is a complex trait controlled by few major and many minor effect genes.

## 1. Introduction

Maize (*Zea mays* L.) is an important cereal crop and a major determinant of food security in Sub-Saharan Africa (SSA) [1]. Maize is grown around 25 million hectares in SSA translating to 38 million metric tons of grain yield. However, maize productivity is severely affected by abiotic and biotic factors including drought, diseases, pests, and socio-economic factors [2]. Recently, maize lethal necrosis (MLN) has led to complete yield losses and thus affected food security negatively [3]. In 2011, MLN was first reported in Bomet District in Kenya with 30–100% yield losses and in 2012 similar symptoms were observed in Chepalungu, Narok, and Naivasha districts in Kenya [4]. It was later confirmed in several countries such as Rwanda [5], Ethiopia [1], DRC [6], and Uganda [4].

MLN is a viral disease resulting from the synergistic infection of two viruses, maize chlorotic mottle virus (MCMV) belonging to the Tombusviridae group and any virus from the Potyviridae group, mostly Sugarcane mosaic virus (SCMV) in SSA [4]. Similar to many viral diseases, MLN is mainly spread by insect vectors of the two viruses and with few incidences through infected seed [7]. MCMV is transmitted in a semi persistent manner by beetles and thrips with thrips playing a major role in the movement of MCMV in Africa [1,8] while SCMV by aphids in a non-persistent manner [3].

Many economic traits in maize are controlled by quantitative genes/loci and are affected to genotype × environment interactions. Therefore, the identification of these genomic regions has gained popularity in maize breeding programs [9]. Presently, various next generation genotyping methods have been developed to facilitate the identification of genes by allowing genotyping with high throughput markers [10]. Furthermore, the exploration of the genes associated with the trait of interest through genome wide association studies (GWAS) presents great opportunities for maize breeding. GWAS is an approach used to identify genes and understand the genetic architecture of complex traits by exploiting linkage disequilibrium (LD) resulting from trait and marker associations [11]. GWAS offers numerous advantages compared to the traditional marker assisted selection (MAS) in that it uses natural populations for instance a collection of individual varieties or inbred lines [12]. Thus, it has a power to dissect historical recombinations through LD analysis [13]. This in turn provides a greater ability and resolution to identify favorable genetic loci controlling the trait of interest and an opportunity to analyze the architecture of complex quantitative traits [14]. However, GWAS is prone to false positive quantitative trait locus (QTL) detections that arise from stratification differences in genetic population structure [15]. These type 1 errors can be controlled using Bayesian model-based cluster that was developed to infer population structures in complex pedigree populations and relatedness in association mapping panels [16]. It involves a set of random markers that are used to estimate the population structure (Q) which is then incorporated together with the kinship relations of the sample into a mixed linear model (MLM) to test associations [17]. Besides population structure, LD also determines the resolution of GWAS [13]. In maize, GWAS has been successfully used to understand the genetic architecture of several maize diseases such as gray leaf spot [18], fusarium ear rot [19], MLN [15], and SCMV [20]. These studies have shown the utility of GWAS in identifying genes controlling the traits of interest.

A vast number of molecular markers are available allowing breeders to use in plant breeding programs [21]. Genomic prediction (GP) contrary to the traditional MAS utilizes genome-wide markers to estimate the effects of all loci and thereby compute a genomic estimated breeding value (GEBVs) [22]. Complexity of most of economically important traits in maize makes the use of genome-wide markers in GP more effective and comprehensive in their improvement [23]. Various approaches utilizing GP have been proposed; including ridge regression-BLUPs [24], Bayesian methods, and machine regression [25]. Among different biometrical approaches in plant breeding, RR-BLUPs commonly used to predict the performance of the unphenotyped lines for complex traits [25]. GP has been applied in maize for important diseases like MLN [15], MCMV [26], northern leaf blight [27], and fusarium ear rot [19]. These studies showed the potential of GP in improving resistance to important maize diseases.

In the present study we used a set of 1400 diverse maize tropical inbred lines to evaluate their response to MLN under artificial inoculations and genotype them with genotyping by sequencing (GBS) to apply GWAS and GP. The objectives of this study were to (1) evaluate large diverse array of tropical and sub-tropical lines to assess their response of MLN under artificial inoculation, (2) validate earlier findings and further identify new marker-trait associations using GWAS, and (3) assess the potential of GP for MLN resistance in maize.

## 2. Materials and Methods

### 2.1. Plant Materials and Trial Design

Fourteen hundred maize inbred lines developed either through pedigree breeding and doubled haploids (DH) technology under IMAS (Improved Maize for African Soils), DTMA (Drought Tolerant Maize for Africa), and WEMA (Water Efficient Maize for Africa) projects at the International Maize and Wheat Improvement Center (CIMMYT) were used. The inbred lines were evaluated in one-row 3-m plots with two replicates in alpha lattice design in two seasons at MLN Screening Facility at the Kenya Agriculture and Livestock Research Organization (KALRO) center at Naivasha (Latitude 0°43′ S, longitude 36°26′ E, 1896 asl), Kenya. The subset of lines from IMAS and DTMA panels were used in our earlier studies [15,26] however, lines from WEMA panel and all lines from IMAS and DTMA panel are additionally included in this study. Two seeds were planted per hill and thinned three weeks after emergence to one plant per hill in order to ensure a uniform number of plants per entry. All standard agronomic practices were followed.

### 2.2. Viral Inoculum and Artificial Inoculation

The SCMV and MCMV isolates collected from MLN hotspot areas in Kenya were used to develop inoculum for this study. The isolates were confirmed by enzyme-linked immunosorbent assay (ELISA). Maintenance of inoculum purity was carried out on the susceptible hybrid H614 in separate greenhouses until artificial inoculation of entries in the field. MLN inoculum was prepared from an optimized combination of MCMV and SCMV viruses in the ratio of 1:4 [1,15] to ensure uniform MLN pressure in the experiment. Plant leaves used for inoculum were cut into small pieces and ground in 10 mM potassium phosphate at pH 7.0. The resulting sap extract was centrifuged at 12,000 rpm for two minutes and decanted with carborundum at 0.02 g/mL. Plants were inoculated twice at an inoculation spray pressure of 10 kg/cm^2^ using a backpack mist blower with an open nozzle of 2 inches in diameter. The presence of both viruses (SCMV and MCMV) in the inoculated field trials was confirmed by ELISA at approximately two weeks after inoculation. Disease severity (DS) was visually scored three weeks after second inoculation at 10-day intervals with four observations made. The scoring was done on a scale of 1 (resistance, no disease symptoms) to 9 (highly susceptible, complete plant death). After analyzing DS for each time score, we selected the third score (40 days post-inoculation) for further analysis because of its higher heritability estimate and full expression of disease symptoms. Area under the disease progress curve (AUDPC) was calculated for each plot to provide a measure of the progression of MLN severity across time.

### 2.3. Phenotypic and Genotypic Data Analysis

Analysis of variance was calculated for DS and AUDPC within and across environments for inbred lines by restricted maximum likelihood method using META-R (Multi Environment Trait Analysis R software). The mixed model used to estimate the variance components was:
MLN ~ Replications (Environment) + Environment + Lines + Lines × Environment + Blocks (Environment). 


Seasons were treated as environments. Broad sense heritability was estimated as the ratio of genotypic to phenotypic variance. Best linear unbiased estimate (BLUE) and best linear unbiased predictor (BLUP) for each genotype were obtained for within and across environments.

Out of the 1400 lines evaluated in the field, 915 lines were genotyped with GBS. DNA of all 915 inbred lines was extracted from 3–4 leaves stage seedlings and genotyped using Genotyping by Sequencing platform at the Institute for Genomic Diversity, Cornell University, Ithaca, USA as per the procedure described in earlier studies [28]. The ~955K GBS SNP datasets were filtered where a minor allele frequency of <0.05, heterozygosity of >5% and missing data rates >10% were excluded from further analysis in TASSEL ver. 5.2.

### 2.4. Population Structure, PCA, and Linkage Disequilibrium Analysis

The LD and principal components (PCs) were calculated using Tassel ver. 5.2. LD decay rate between each pair of SNPs was analyzed with the squared Pearson correlation coefficient (r^2^). The rate of LD decay with physical distance was visualized and average pairwise distances at which LD decayed at r^2^ = 0.1 and 0.2 were calculated in R software. Structure V2.3.4 software was used to divide the population into subgroups using 5085 SNPs (MAF ≥ 0.4) and 915 maize inbred lines. The population structure of the studied maize panel was further investigated by the STRUCTURE software. The population structure identified from STRUCTURE software was subjected to STRUCTURE HARVESTER to identify Delta K values. The reasonable subgroups number (K) was obtained using Delta K value from Structure Harvester. We used K values ranging from 1 to 10. The strong Markov Chain Monte Carlo (MCMC) after the non-repeated iteration was set to 10,000 times at the beginning and then set to 10,000 times with the number of iterations set at 2. The PCs and categorical data were plotted by CurlyWhirly v1.15 (http://ics.hutton.ac.uk/curlywhirly/) and R software to obtain the explained variance of each PCs.

The BLUP values obtained for DS and AUDPC were used in GWAS as phenotypes. The kinship matrix obtained with a centered identity by state (IBS) and the first five PCs which explained maximum variation were used to correct the population structure in a mixed linear model using TASSEL ver. 5.2 [29]. Genome wide scans for marker-trait associations were conducted with mixed linear model. The amount of phenotypic variation explained by the model was assessed using the R2 statistics, calculated by fitting all significant SNPs simultaneously in a linear model in R. Multiple testing correction was performed by using false discovery rate (FDR *p* < 0.01) and significant associations were declared when the *p*-values are less than 1 × 10^−6^. The 50 bp source sequences of the significantly associated SNPs were used to perform BLAST searches against the B73 RefGen v2 genome set in Maize GDB (http://www.maizegdb.org). The putative candidate genes identified in MaizeGDB were within or adjacent to each associated SNP.

### 2.5. Genomic Prediction

GP was carried out with RR-BLUP [30] with five-fold cross-validation. BLUEs across environments were used for this analysis for DS and AUDPC. So as to examine the effects of training population and marker density on genomic prediction accuracy, we used five levels of training population size (i.e., 230, 456, 685, and 915) with 6300 SNPs and five levels of marker density (i.e., 500, 1000, 2000, 4000, and 6300 SNPs) with 915 lines to evaluate the prediction accuracy of each trait. Combined population prediction approach where all panels are combined and randomly sampled to form both a training set and a testing set was used. The prediction accuracy was obtained as the correlation between GEBVs and the observed phenotypes divided by the square root of the estimated heritability. Sampling of the training and validation sets were repeated 100 times.

## 3. Results

### 3.1. Phenotypic Analysis

ANOVA across environments revealed significant genotypic variance for both MLN-DS and MLN-AUDPC (Table 1). For both MLN-DS and AUDPC values, GxE interaction variances were significant at *p* < 0.05. This ruled out the possibility of bias due to environment-specific disease responses in a combined analysis. The frequency of the phenotypic values in both MLN-DS and AUDPC revealed an approximately normal distribution (Figure 1). Across environments, heritability was moderate for MLN-DS with 0.42 and high for AUDPC with 0.86.

### 3.2. PCA and Population Structure

The first three PCs explained about 72% of the total variance. PC1 and PC2 explained 32% and 22% of the total variance, respectively (Figure 2a). We identified Delta K values because LnP (D) was continuously increasing with the K value and was not best for identifying the groups. The line plot for Delta K values suggested that the population could be structured in to two or three groups in order of possibility as shown by the peaks (Figure 2b,c). In the whole set of panels, lines derived from IMAS project were distributed along with group of lines derived from DTMA and WEMA panel unlike DTMA panel which was more structured (Figure 2c).

### 3.3. Linkage Disequilibrium and GWAS

LD of the whole genome was estimated using 49,547 SNPs (MAF ≥ 0.05). The genome-wide LD decay was plotted as LD (r^2^) between adjacent pair of markers versus distance in kb (Figure 3). Results showed that LD decayed differently in the physical distance. A rapid decline in LD was observed with increasing physical distance. At a cut-off value of r^2^ = 0.1 and 0.2, the average physical distance was 1.78 kb and 0.62 kb, respectively.

From the GBS data, a set of 215,137 high-quality SNPs were retained for GWAS. The GWAS results for MLN-DS and AUDPC across environments are shown in Manhattan plots and Q-Q plots of P values comparing expected −log10 p value to observed −log10 p value in Figure 4. We detected 32 significantly associated SNPs for MLN resistance whereby 5 SNPs overlapped between the two traits (Table 2, *p* < 1.0 × 10^−6^). For MLN-DS, 18 significantly associated SNPs individually explained 3–5% of the total phenotypic variance and together explained 17.05% of the total phenotypic variance. Whereas for AUDPC, 19 significantly associated SNPs individually explained 3–12% of the total proportion of phenotypic variance and together explained 24.5% of the total phenotypic variance. Among these significantly associated SNPs, S1_44539940 on chromosome 1 and S4_199711804 on chromosome 4 are found to be the most significantly associated SNPs for DS and AUDPC, respectively. Among several genomic regions identified for MLN resistance, allelic effects on MLN resistance was prominent in selected eight SNPs. The phenotypic values of the different allele classes of these SNPs in association panel for MLN-DS and AUDPC value were presented in Figure 5. A set of putative candidate genes associated with the significant markers were identified and their functions revealed, they were directly or indirectly involved in plant defense responses (Table 2).

### 3.4. Genomic Prediction

Prediction accuracies generated for MLN-DS and AUDPC are shown in Figure 6. GP accuracy was increased gradually with increasing in marker density from 500 to 6300. When training population size (TPS) was 915, the mean prediction accuracy of MLN-DS was 0.60 with marker density of 500; 0.64 with marker density of 1500; 0.68 with marker density of 3000; 0.71 with marker density of 4500; and 0.72 with marker density of 6,300. The mean prediction accuracy for AUDPC at 500, 1500, 3000, 4500, and 6300 SNPs were 0.36, 0.42, 0.45, 0.46, and 0.50, respectively. Increase in training population size from 230 to 915 showed slight increase in the prediction accuracies at constant marker density. When TPS was 915, 685, 456, and 230 lines, the mean prediction accuracy was 0.72, 0.61, 0.59, and 0.57, respectively for MLN-DS and 0.50, 0.48, 0.45, and 0.36, respectively for AUDPC values.

## 4. Discussion and Conclusions

For the last six years, MLN has been recognized as one of the major diseases constraining yield production in SSA. Thus, effective breeding strategies for MLN resistant germplasm have to be employed [3]. In this study, we phenotyped and genotyped a large set of diverse maize lines and applied GWAS and GP in order to understand the genetic architecture of MLN resistance, validate previous findings in subtropical germplasm and analyze population structure. We also reported the results of the analysis of the population structure and LD patterns in the used maize germplasm panel.

Results in this study showed that the genetic variance for both the traits were significant and the heritability was moderate to high. This information was consistent with previous studies on MLN resistance which observed significant genetic variances and moderate to high heritability [2,15,26,31]. Higher estimates of heritabilities depict the ability of the traits for improvement of the maize germplasm for MLN resistance. The high quality of the phenotypic data makes it more suitable for association mapping studies [17].

For GWAS, MLM has been used successfully in multiple populations for correcting population stratification using kinship matrix [32]. The resolution and the required marker density of GWAS depends on the extent of LD in the population [33]. Our study LD was decayed rapidly and at cut off points of r^2^ = 0.1 and 0.2 the distance was 1.78 kb and 0.62 kb, respectively thus suggesting the potential of GWAS for this study. Previous studies showed that GBS and the association power was enhanced when LD decay distance throughout all the 10 chromosomes in an association mapping panel was less than 5 kb at r^2^ = 0.1 [34,35,36]. The observed LD decay was also in line with the LD decay observed on tropical maize panels used for haploid male fertility study [37].

The line plot for Delta K values suggested that the population could be structured into two or three groups. The three groups could be associated with the three panels we used. These association panels represent breeding programs from across Africa and Latin America posing the possibility of false positives arising from population structure. Therefore, we used the first five PCs with relative kinship matrix to correct for false associations.

The genetic architecture of MLN has been reported in biparental and association panels [15,26,31]. The present study showed that GWAS analyses is powerful for revealing marker-trait associations for both MLN-DS and AUDPC using high marker density. We identified 32 SNPs significantly associated with these two traits using GWAS with 18 SNPs were for MLN-DS and 19 SNPs for AUDPC. For MLN-DS, these SNPs were distributed across all chromosomes (chr) except chr 2, 5, 6, and 8. Five SNPs are common whereas 13 and 14 SNPs are specific for MLN-DS and AUDPC, respectively. These results suggest that in the used panel, MLN is controlled by minor genes distributed across the genome and thus explaining that MLN is complex in nature. We identified a genomic region S1_44539940 and S4_199711804 are valuable for both the traits. Ten SNPs (S3_33757503, S3_55239348, S3_56468811, S3_136082606, S3_147938951, S3_149313702, S3_161574458, S3_161574468, S3_161574470, S3_161574471, and S3_190890553) were detected on chromosome 3 either for MLN-DS and/or AUDPC localized to the map bins 3.04 and 3.05 corroborating with a previous study on MLN [16].

Among the 32 SNPs identified together for MLN disease severity and AUDPC values, most were overlapped with SNPs reported in earlier three studies where multiple populations were used with linkage mapping, joint linkage association mapping (JLAM), and GWAS approaches [15,26,31]. SNP S4_199711804 is another significantly associated SNP found in this study is falling within the confidence interval of major effect QTL, qMLN4-235 reported for MLN [26]. Among 10 SNPs found on chromosome 3, S3_55239348, S3_56468811 were coincided with the QTL reported for both MCMV and MLN [26], whereas SNPs S3_136082606, S3_147938951, S3_149313702, were coincided with QTL reported for MLN in four biparental populations [31]. Four SNPs, S3_161574458, S3_161574468, S3_161574470, and S3_161574471 are overlapping with the QTL reported for MLN in three DH populations [26]. Interestingly SNP S3_190890553 was falling very close to the MCMV and MLN resistance associated QTL qMCMV2-189, identified through both GWAS [15] and JLAM across three DH populations [26].

Nevertheless, the SNP identified on chromosome 6, S6_148513637 is falling very close to the consistent QTL qMLN6-158 reported for both MCMV [26], and MLN [31]. Interestingly, a QTL for SCMV resistance is very close region from diversity panel [36]. Another marker S6_99770682 is overlapped with QTL qMLN6-100 reported through JLAM and located very close to the QTL qMLN6-89 reported in biparental populations [31]. Two SNPs on chromosome 7, S7_140411743 and S7_143109798 are located within the QTL qMLN7-144 which has confidence interval of 139 to 149 Mbp reported in biparental populations [26]. Other SNP on chromosome 7 S7_166270242 is located within the QTL qMLN7_152 which had confidence interval of 152 to 167 Mbp [26]. Another SNP detected on chromosome 8, S8_150798179 is falling within the confidence interval of the QTL detected for MCMV in DH population [26]. Three SNPs located closely on chromosome 10 even though detected as new markers but located very closely to the QTL qMLN10_137 detected across DH populations through JLAM [26]. The results of the present study help to reduce the detected QTLs big confidence interval and helps to focus on few markers or specific region which can enhance the efficiency of markers in improving MLN resistance. The allelic effect of some of these markers clearly support the effective role in improving the level of MLN resistance (Figure 5). These consistent regions or SNPs could potentially help the breeders to design effective strategy to introgress these QTL in relevant breeding materials through marker assisted breeding.

In the previous MLN studies, a set of putative candidate genes significantly associated with MLN were identified, i.e., candidate gene GRMZM2G109805 on chr 5 involved in hypersensitivity reaction, GRMZM2G018943 that functions as a translation initiation factor eIF-2B involved in mutations in host factors, GRMZM2G056612 and GRMZM2G008109 involved in protein serine/threonine kinase activity that plays a role in cell signaling for pathogen perception and plant defense activation [15]. In addition, Gowda et al. [15], reported two putative candidate genes namely GRMZM2G471517 and GRMZM2G404316 that are antifreeze pathogenesis-related proteins with ice binding properties. In the present study, GWAS revealed a set of putative candidate genes some of which are involved in plant defense, cell development, and signaling. The putative gene GRMZM2G159402 is annotated as being involved in RNA biosynthesis and transcriptional activation for tissue specific expression in maize [38]. We also annotated GRMZM2G024159 which is one of the housekeeping genes in maize primarily involved in maintaining basic cellular functions [39].

Another putative candidate gene GRMZM2G071015 encodes for a BAG-associated GRAM protein and is involved at the most elementary level for modifying and controlling maize leaf lipidomes which could play a role as an indicator for respective stress [37]. GRMZM2G428168, is involved in protein modification, and S-glutathionylation and deglutathionylation. It is involved in maize glutathione-S-tranferase (GST) protein expression which plays a role in plant defense for instance biotic stress [40]. Numerous studies have shown the induction of GSTs in early stages of bacterial, fungal, and viral infections and that silencing of these GSTs could modify disease symptoms and multiplication rates [41]. This putative candidate gene could play a role in MLN resistance as these GSTs can detoxify toxic substances that accumulate during infection by their conjugation with glutathione and attenuation of oxidative stress [41]; however, more research is needed to reveal the exact relevance of the GST gene in MLN resistance.

The candidate gene GRMZM2G177046 is involved in RNA biosynthesis and basic-region leucine zipper (bZIP) transcription factor activation that regulate plant growth and development and are also involved in stress responses and hormone signaling [42]. It is an ocs element binding (OBF) factor 1 and notably, OBFs are a group of promoter sequences belonging to a very specific class of bZIP transcription factors required for the expression of both genes of pathogens in infected plants and plant defense genes [43]. The OBFs bind to a family of related, 20-bp DNA promoter sequences called ocs elements which bacterial and viral pathogens use to express genes in plants [43]. Moreover, the ocs elements also regulate the transcription of GSTs which involved in plant defense responses [41].

Mitogen activated protein kinase (MAPK) cascades are conserved signal transduction pathways that translate external stimuli into cellular responses in all eukaryotes [44]. The MAPK kinase kinases (MAPKKK) are activated by upstream signals to phosphorylate a MAPK kinase (MKK) that in turn activated a specific MAPK. MAPK pathways are activated in signaling pathways such as plant innate immunity, plant defense, and hormone responses [43,45]. GRMZM2G400470 is a MAPKK2 which was reported to be activated in an MAPK cascade namely MEKK1-MKK1/MKK2-MPK4 in *Arabidopsis* during plant pathogen perception [46]. It was also induced upon *Fusarrium verticilliodes* inoculation in a CO441 maize genotype thus explaining its importance in resistance in the particular genotype to *F*. *verticilliodes* that causes ear rot and accumulation of mycotoxins [46]. Furthermore, we also found another putative candidate gene GRMZM2G409309 which is associated with a powdery mildew resistant protein 5/trichome Birefringence-like 38 and has also shown to be involved in the partial resistance in maize common rust caused by the fungus *Puccinia sorghi*. In *Arabidopsis*, PMD5 mutant lines have cell walls rich in pectin and increases résistance to the powdery mildew pathogens *Erisyphe cichoracearum* and *E*. *orontii* and further studies have suggested its importance in non-host resistance in the crop [46,47].

Unlike traditional MAS, GP incorporates all available genome-wide markers thus capturing all major and minor marker effects. GP has been applied successfully in previous studies in maize with moderate to high prediction accuracies [27,35,48] making it an effective and powerful approach for complex trait improvement. In the present study, GP revealed moderate to high prediction accuracies for MLN-DS and AUDPC (Figure 6) that supported their quantitative nature. These results were comparable with earlier studies for MLN in which Gowda et al. [15] and Sitonik et al. [26] reported moderate to high prediction accuracies. The results also showed that the prediction accuracies were significantly associated with marker density which also agreed with previous studies whereby the prediction accuracies increased with increase in marker density [49]. This indicates that higher marker density is required to obtain higher prediction accuracies for complex traits. However, the tradeoff between cost of high-density genotyping and gain in prediction accuracy be considered carefully for breeding application. Interestingly varying training population size did not show big differences in prediction accuracies possibly because of the different training population sets used had more information as compared to others. MLN-DS showed a higher prediction accuracy compared to AUDPC possibly because it is less complex compared to AUDPC. Gowda et al. [31], reported that QTL mapping and joint linkage association mapping (JLAM) results suggested that the genetic architecture of MLN was possibly less complex compared to traits such as grain yield while the complexity of MLN was showed more from the GP results. Therefore, it is important to incorporate GP in breeding programs because it allows the capture both small effect QTLs together with major effect QTLs [11].

In conclusion, we used a large set of lines together comprising 1400 inbreds to understand and validate the genetic architecture of MLN resistance and identified 32 SNPs significantly associated with MLN resistance. The GP results revealed that it can be used to improve resistance to MLN and also to predict germplasm response to MLN. However, more research is necessary to validate the identified candidate genes and their functions to relate specifically to MLN resistance.

## Figures and Tables

**Figure 1 genes-11-00016-f001:**
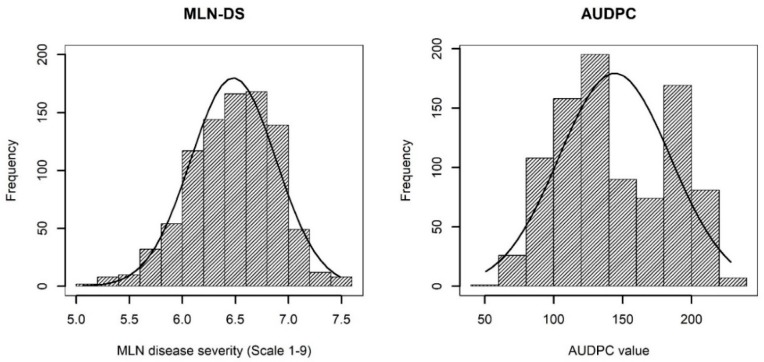
Phenotypic distribution of disease severity (MLN-DS) on the scale of 1–9 and the area under disease progress curve (AUDPC) values for maize lethal necrosis (MLN)**.**

**Figure 2 genes-11-00016-f002:**
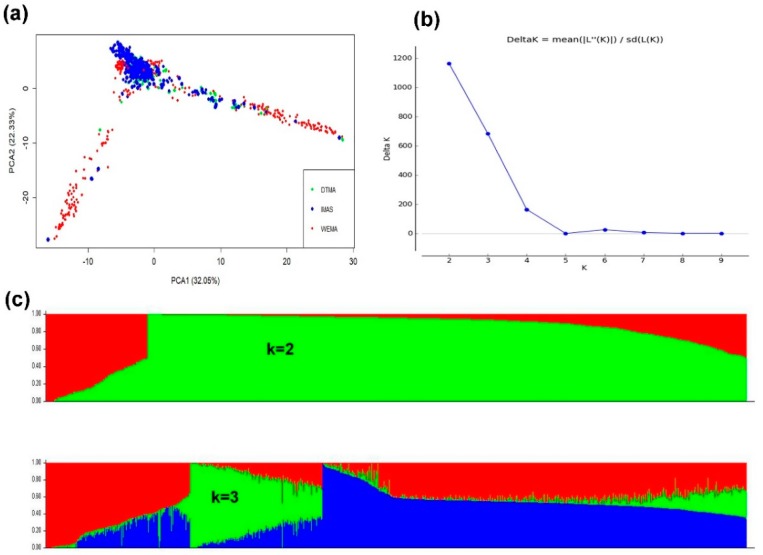
Population structure and PCA plot of 914 maize inbred lines estimated from 5085 SNPs. (**a**) PCA plot for the entire population and colored by the group divisions (DTMA, IMAS, and WEMA). (**b**) Plot of Delta K was calculated for K = 2 to K = 9. (**c**) Population structure of the lines for K = 2 and K = 3.

**Figure 3 genes-11-00016-f003:**
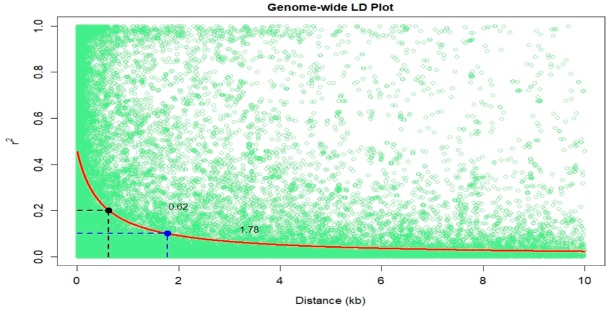
Linkage disequilibrium (LD) plot representing the average genome-wide LD decay in the panels with genome-wide markers. The values on the *y*-axis represents the squared correlation coefficient r^2^ and the *x*-axis represents the physical distance in (kb).

**Figure 4 genes-11-00016-f004:**
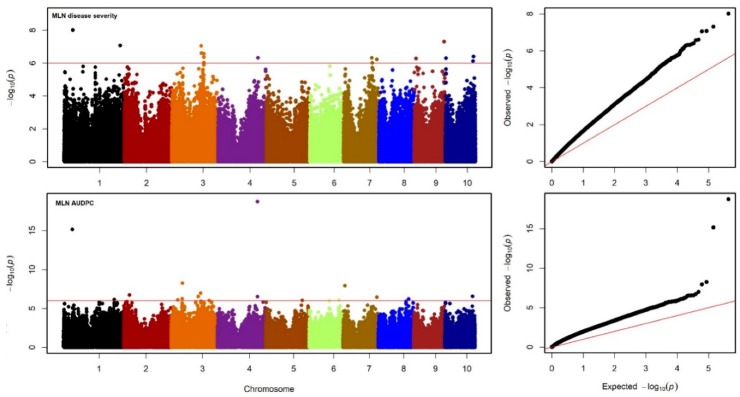
Manhattan and Q-Q plots for the GWAS of MLN for disease severity and the AUDPC value. The dashed horizontal line in Manhattan plots depicts the significance threshold (*p* = 1 × 10^−7^). The *x*-axis indicates the SNP location along the 10 chromosomes, separated by different colors. the red line in the Q-Q plots depicts the line of best fit whereby for both traits the plot of expected −log_10_(*p*) against observed −log_10_(*p*) falls above the line of best fir.

**Figure 5 genes-11-00016-f005:**
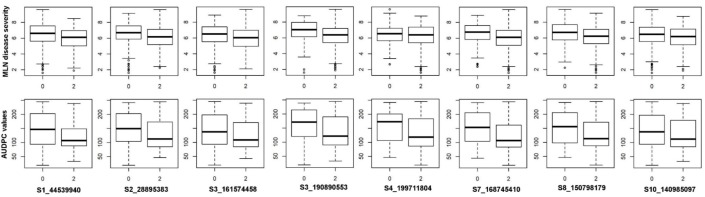
Box plots showing the phenotypic values of the different allele classes of eight SNPs identified in GWAS for MLN disease severity and AUDPC value. The SNP names and alleles are mentioned below. The black horizontal lines in the middle of the boxes are the median values for the MLN disease severity and AUDPC value in the respective allele classes.

**Figure 6 genes-11-00016-f006:**
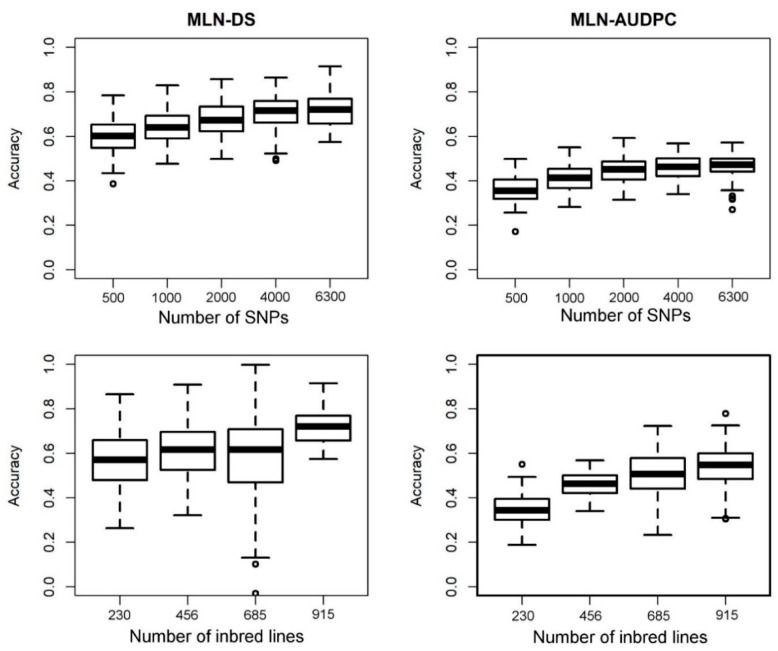
Effect of the number of markers and number of individuals on the accuracy of genomic prediction when the number of markers and size of the training population varied for MLN-DS and AUDPC.

**Table 1 genes-11-00016-t001:** Mean, range, and variance components for disease severity (DS) and AUDPC for maize inbred line association mapping panel

	DS (1–9)	AUDPC
Mean	6.25	135.28
Minimum	5.30	58.69
Maximum	7.12	216.05
LSD	0.93	20.43
σ^2^_G_	0.55 **	2303.16 **
σ^2^_GxE_	0.78 **	143.36 *
σ^2^_e_	1.40	1267.99
*h* ^2^	0.42	0.86

*^,^** Significance at *p* < 0.05 and *p* < 0.01, respectively. σ^2^_G,_ σ^2^_GxE_ and σ^2^_e_, represents variance at genotypic, genotype x environment interactions and error, respectively, h^2^—heritability.

**Table 2 genes-11-00016-t002:** Details of the MLN resistance associated SNP markers and their probable candidate genes identified in the large set of association mapping panel.

SNP	Chr	Position (bp)	MLM P-Values	R^2^	MAF *	Allele	MAE **	Putative Candidate Genes	Predicted Function of Candidate Gene
MLN disease severity									
S1_44539940	1	44539940	9.66E-09	0.05	0.10	C/T	0.42	GRMZM2G024159	protein YIPF5 homolog
S1_281333891	1	281333891	8.46E-08	0.04	0.17	C/G	0.07	GRMZM2G177046	bZIP transcription factor
S3_147938951	3	147938951	8.93E-08	0.04	0.12	C/G	−0.22	GRMZM2G044867	unknown
S3_149313702	3	149313702	2.50E-07	0.04	0.22	A/G	−0.02	GRMZM2G428168	S-glutathionylation and deglutathionylation
S3_161574458	3	161574458	9.25E-07	0.04	0.09	C/G	−1.97	GRMZM2G145346	PAK-box/P21-Rho-binding
S3_161574468	3	161574468	4.40E-07	0.03	0.11	T/A	0.73	GRMZM2G145346	PAK-box/P21-Rho-binding
S3_161574470	3	161574470	1.25E-06	0.03	0.09	A/G	0.75	GRMZM2G145346	PAK-box/P21-Rho-binding
S3_161574471	3	161574471	2.69E-07	0.04	0.09	C/A	0.8	GRMZM2G145346	PAK-box/P21-Rho-binding
S4_199711804	4	199711804	4.76E-07	0.04	0.29	C/T	−0.11	GRMZM2G134857	uncharacterized protein
S7_140411743	7	140411743	4.84E-07	0.04	0.07	C/T	1.49	GRMZM2G071015	BAG-associated GRAM protein
S7_143109798	7	143109798	9.81E-07	0.03	0.37	C/T	0.71	GRMZM2G179021	RNA.regulation of transcription.
S7_166270242	7	166270242	5.95E-07	0.04	0.11	G/C	0.54	GRMZM2G520980	unknown
S9_9599125	9	9599125	5.21E-07	0.03	0.08	A/C	1.25	GRMZM2G159402	transcriptional activation
S9_149758216	9	149758216	4.94E-08	0.04	0.25	T/C	0.11	GRMZM2G540298	unknown
S10_3189860	10	3189860	4.97E-07	0.04	0.05	A/G	−1.16	GRMZM5G862857	uncharacterized protein
S10_138075442	10	138075442	7.56E-07	0.03	0.26	G/C	0.77	GRMZM2G117667	GDSL-like Lipase/Acylhydrolase superfamily protein
S10_138075445	10	138075445	7.56E-07	0.03	0.26	C/T	0.77	GRMZM2G117668	unknown
S10_140985097	10	140985097	3.94E-07	0.04	0.08	T/C	0.06	GRMZM2G109753	scramblase family protein
**Total R^2^**				**17.05**					
**Area under disease progress curve**									
S1_44539940	1	44539940	6.92E-16	0.1	0.1	C/T	10.15	GRMZM2G024159	Yip1 domain containing protein
S1_253798682	1	253798682	7.08E-07	0.03	0.33	G/A	23.06	GRMZM2G043127	translocase of the outer mitochondrial membrane
S2_28895383	2	28895383	1.86E-07	0.03	0.42	A/G	7.3	GRMZM2G077420	unknown
S3_33757503	3	33757503	7.85E-07	0.03	0.13	T/C	−18.42	GRMZM2G563119	unknown
S3_55239348	3	55239348	5.33E-07	0.03	0.37	C/G	4.69	GRMZM2G520940	protein coding
S3_56468811	3	56468811	5.63E-09	0.04	0.43	A/G	3.02	GRMZM2G409309	powdery mildew resistant protein5
S3_136082606	3	136082606	2.87E-07	0.04	0.39	G/C	32.3	GRMZM2G092169	uncharacterized protein
S3_147938951	3	147938951	1.04E-07	0.04	0.12	C/G	−0.22	GRMZM2G044867	unknown
S3_190890553	3	190890553	9.37E-07	0.03	0.15	G/A	1.6	GRMZM2G563190	mitochondrial electron transport/ATP synthesis.
S4_199711804	4	199711804	1.89E-19	0.12	0.29	C/T	−2.87	GRMZM2G134857	uncharacterized protein
S4_200034077	4	200034077	3.06E-07	0.03	0.16	A/G	12.07	GRMZM2G465165	ATP binding/amino acid phosphorylation
S5_182091386	5	182091386	8.77E-07	0.03	0.25	T/A	39.18	GRMZM2G137426	protein dimerization activity
S6_99770682	6	99770682	1.04E-06	0.03	0.39	T/G	22.51	GRMZM2G112337	MAP65-2 microtubule-associated protein
S6_148513637	6	148513637	8.16E-07	0.03	0.07	A/G	47.5	GRMZM2G020856	O-fucosyltransferase family protein
S7_8677545	7	8677545	1.16E-08	0.05	0.24	C/G	−10.84	GRMZM2G107408	uncharacterized protein
S7_168745410	7	168745410	3.55E-07	0.03	0.4	A/G	1.59	GRMZM2G039757	tolB protein-related
S8_150798179	8	150798179	5.92E-07	0.03	0.43	A/C	−7.05	GRMZM2G531490	unknown
S10_138075442	10	138075442	2.75E-07	0.03	0.26	G/C	0.77	GRMZM2G117667	GDSL-like Lipase/Acylhydrolase superfamily protein
S10_138075445	10	138075445	2.75E-07	0.03	0.26	C/T	0.77	GRMZM2G117668	unknown
**Total R^2^**				**24.75**					

* is minor allele frequency, ** is minor allele effect, and MLM is mixed linear mod.

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
