# Peer review of "Genome-Wide Analyses and Prediction of Resistance to MLN in Large Tropical Maize Germplasm"

_genes, 2019, doi:10.3390/genes11010016_

Round 1

Reviewer 1 Report

General

This manuscript applied genome wide markers to predict genetic regions that might be involved in Maize lethal necrosis (MLN) on a large set of inbred lines. Writing of this manuscript is clear and straightforward. It offers a scalable system to identify candidate genes involve in the disease. If adds an extra step on previous studies in capturing both the small effects QTLs together with the major effect QTLs. It also validates and fine map the QTL regions into specific regions that pinpoint genes involve in plant defense, cell development and signaling. All these candidates open a great opportunity to understand their putative relationship to MLN resistance. Overall it is a great study that provide a good approach to identify resistance genes that can be used in breeding programs.

Minor comments

Lines 53-54: “Furthermore, the exploration of adjacent genes through genome wide association studies (GWAS) presents for maize breeding.” Not sure I understand the sentence. Does it want to say that the exploration of those genes presents great opportunities for maize breeding? Missing something.

Line 129: “Out of the 1,400 lines evaluated in the field, 915 lines were genotyped” What happened to the 485 lines that were not genotyped? Why were they excluded?

Line 154: “the significantly associated SNPs were used to perform BLAST searches against the B73 genome set”. Table 2 listed the IDs corresponding to B73 RefGen v3. Maybe mention it on the material and method section, some maize researchers might be using different annotation version. 

Line 168: The phenotypic analysis was made in the 1,400 lines but the genotypic analysis was made on 915. Will that make a difference in the overall analysis? Should the 485 lines that were not genotyped removed from the study? I imagine both parameters can be analyzed independently, but I just want to be sure about the impact of having data on phenotypes that did not get genotyped.

Line 196: Figure 2. Label a, b, c on the plots and graph. Maybe mention in the figure that color schemes reflect difference in environment (soil, drought tolerance and water efficiency).

Line 208: “32 significantly associated SNPs for MLN resistance”. For MLN-DS, there 18 and for AUDPC, 19. Maybe add that 5 of the candidate genes overlapped. It is mentioned in the discussion section so it is not critical.

Line 241: Figure 4. Explain Q-Q plots results while there is a description of the Manhattan plots in the Figure, I don’t see any explanation in the results or discussion section.

Author Response

Comments and Suggestions for Authors

This manuscript applied genome wide markers to predict genetic regions that might be involved in Maize lethal necrosis (MLN) on a large set of inbred lines. Writing of this manuscript is clear and straightforward. It offers a scalable system to identify candidate genes involve in the disease. If adds an extra step on previous studies in capturing both the small effects QTLs together with the major effect QTLs. It also validates and fine map the QTL regions into specific regions that pinpoint genes involve in plant defense, cell development and signaling. All these candidates open a great opportunity to understand their putative relationship to MLN resistance. Overall it is a great study that provide a good approach to identify resistance genes that can be used in breeding programs.

Response: Thanks for the constructive comments on the study

Comment 1

Lines 53-54: “Furthermore, the exploration of adjacent genes through genome wide association studies (GWAS) presents for maize breeding.” Not sure I understand the sentence. Does it want to say that the exploration of those genes presents great opportunities for maize breeding? Missing something.

Response: We modified the specific sentence to make it clearer and more meaningful in the revised manuscript. Please see Line 54-55

Comment 2

Line 129: “Out of the 1,400 lines evaluated in the field, 915 lines were genotyped” What happened to the 485 lines that were not genotyped? Why were they excluded?

Response: Thanks for the comment. We genotyped large number of lines for breeding of drought tolerance. When MLN appeared in 2013, we phenotyped a large number of lines and among the phenotyped lines we checked how many had both phenotypic and genotypic data and were used in GWAS. The genotyping of lines was purely random, there was no selection applied.

Comment 3

Line 154: “the significantly associated SNPs were used to perform BLAST searches against the B73 genome set”. Table 2 listed the IDs corresponding to B73 RefGen v3. Maybe mention it on the material and method section, some maize researchers might be using different annotation version. 

Response: Thank you for your suggestion. We included the relevant information in the revised manuscript. Please see Line 161

Comment 4

Line 168: The phenotypic analysis was made in the 1,400 lines but the genotypic analysis was made on 915. Will that make a difference in the overall analysis? Should the 485 lines that were not genotyped removed from the study? I imagine both parameters can be analyzed independently, but I just want to be sure about the impact of having data on phenotypes that did not get genotyped.

Response: Thank you for your comment. We resolved this issue under comment 2. In brief, yes, we did the analyses with all lines 1400 lines for pheno data and 915 lines for GWAS analyses as well as only 915 lines for both pheno and geno analyses. There was not much difference in the analyses results. Since theoretically whether we use all lines or part of lines, variance components will remains same, therefore we decided to provide phenotypic analyses results based on 1400 lines and GWAS for 915 lines.

Comment 5

Line 196: Figure 2. Label a, b, c on the plots and graph. Maybe mention in the figure that color schemes reflect difference in environment (soil, drought tolerance and water efficiency).

Response: Thank you for your suggestion. We label the figures in Figure 2 and indicated what the different color schemes represent in PCA. We believe that the groups could be classified into the different panels that we used i.e. DTMA, IMAS and WEMA.

Comment 6

Line 208: “32 significantly associated SNPs for MLN resistance”. For MLN-DS, there 18 and for AUDPC, 19. Maybe add that 5 of the candidate genes overlapped. It is mentioned in the discussion section so it is not critical.

Response: Thank you for your suggestion. We modified this in the revised manuscript, Line 220-221.

Comment 7

Line 241: Figure 4. Explain Q-Q plots results while there is a description of the Manhattan plots in the Figure, I don’t see any explanation in the results or discussion section.

Response: Sorry for the mistake. We included the description of the Q-Q plots results in the Figure 4. The explanation of the Manhattan and Q-Q plots is under GWAS results and discussion showing the significant SNPs in the 10 chromosomes and where we corrected for population structure.

Reviewer 2 Report

Page 1, line 34. Zea mays must be in italics Page 1, line 37-39. I suggest not using the "100%" word; what about something like "complete lose" or so. It is sounds weird because we know that the losses can not be larger than 100%. Page 1, line 44. Maybe a "," is needed between group" and "mostly". Page 1, line 44 until the end of that paragraph. You used "transmitted" three times and "mainly transmitted" two times. Please use a different word. Page 3, line 107. Is there a reference for the 1:4 ratio used in the study? Page 4, line 177. I suggest removing the first sentence. Page 4, first paragraph of the section 3.2. There are multiple sentences here that need to be moved to the methods section. Page 5. Legend of figure 1. A space after "1-9" is needed. Page 6. Figure 2 needs to be separated in individual panels, with better explanation in the legend. Page 7-10. I suggest Figure 3 and 4 appear before table 2. Page 7-8. I suggest using symbols for detailing abbreviations of on the table's header. For example. MAF*, MAE**, etc. Then describe what * and ** at the bottom of the table. Page 7-8. I suggest ordering markers based on physical position Page 9, line 236-238. Reorganize that sentence Page 10, figure 5. Impossible to see the labels in the figure. Fix that please. Page 10, line 256-257. There is no need for that last sentence. Page 10, last two sentences of the second paragraph. These don’t make that much sense, please re write them. Page 10, lines 296-297. Check that sentence. Page 12, line 294. You said “Among the 32 SNPs identified with MLN-DS and AUDPC values”. What do you mean by MLN-DS and AUDPC values? Additional comments: 1. The whole Discussion section requires extensive revision of the writing. 2. Fix resolution in the figures. In some of them, it is impossible to read the labels. 3. The experiment and data analysis were well executed, however, there are problems with the writing that affect the paper considerably.

Author Response

Comment 1

Page 1, line 34. Zea mays must be in italics

Response: Sorry for the mistake. We corrected in the revised manuscript.

Comment 2

Page 1, line 37-39. I suggest not using the "100%" word; what about something like "complete lose" or so. It is sounds weird because we know that the losses cannot be larger than 100%.

Response: Thank you for your suggestion. We modified it to “complete yield losses” in the revised manuscript

Comment 2

Page 1, line 44. Maybe a "," is needed between group" and "mostly".

Response: Sorry for the mistake. We corrected in the revised manuscript

Comment 3

Page 1, line 44 until the end of that paragraph. You used "transmitted" three times and "mainly transmitted" two times. Please use a different word.

Response: Sorry for the mistake. We corrected in the revised manuscript.

Comment 4

Page 3, line 107. Is there a reference for the 1:4 ratio used in the study?

Response: We included the relevant reference for the protocol in the revised manuscript

Comment 5

Page 4, line 177. I suggest removing the first sentence.

Response: As for the suggestion, we deleted the first sentence of the results section in the revised manuscript

Comment 6

Page 4, first paragraph of the section 3.2. There are multiple sentences here that need to be moved to the methods section.

Response: We moved the sentences to methods section (see line 141-143) and retained the explanation for using Delta K in the results section.

Comment 7

Page 5. Legend of figure 1. A space after "1-9" is needed.

Response: Sorry for the mistake. We corrected in the revised manuscript.

Comment 8

Page 6. Figure 2 needs to be separated in individual panels, with better explanation in the legend.

Response: Thank you for your suggestion. We label the figures in Figure 2 and indicated what the different color schemes represent in PCA.

Comment 9

Page 7-10. I suggest Figure 3 and 4 appear before table 2.

Response: As for your suggestion we moved the figure 3 and 4 before Table 2 in the revised manuscript

Comment 10

Page 7-8. I suggest using symbols for detailing abbreviations of on the table's header. For example. MAF*, MAE**, etc. Then describe what * and ** at the bottom of the table.

Response: Thank you for your suggestion. We modified the abbreviations in the revised manuscript.

Comment 11

Page 7-8. I suggest ordering markers based on physical position

Response: As for your suggestion we arranged the table 2 for MLN-DS and AUDPC values

Comment 12

Page 9, line 236-238. Reorganize that sentence.

Response: We reorganized this sentence in the revised manuscript

Comment 13

Page 10, figure 5. Impossible to see the labels in the figure. Fix that please.

Response: Sorry for the mistake. We replaced with better resolution figures.

Comment 14

Page 10, line 256-257. There is no need for that last sentence.

Response: We removed the sentences in the revised manuscript

Comment 15

Page 10, last two sentences of the second paragraph. These don’t make that much sense, please re write them.

Response: We modified the relevant sentence in the revised manuscript.

Comment 16

Page 10, lines 296-297. Check that sentence.

Response: Sorry for the mistake. We modified the sentences in the revised manuscript

Comment 17

Page 12, line 294. You said “Among the 32 SNPs identified with MLN-DS and AUDPC values”. What do you mean by MLN-DS and AUDPC values?

Response: This is indicating the 32 SNPs identified to be significantly associated with MLN-DS and AUDPC values

Comment 18

Additional comments: 1. The whole Discussion section requires extensive revision of the writing. 2. Fix resolution in the figures. In some of them, it is impossible to read the labels. 3. The experiment and data analysis were well executed, however, there are problems with the writing that affect the paper considerably.

Response: Thanks for your constructive comments. We resolved the resolution of the figures and modified the writing.